# DÉJÀQ: OPEN-ENDED EVOLUTION OF DIVERSE, LEARNABLE AND VERIFIABLE PROBLEMS

## ABSTRACT

Recent advances in reasoning models have yielded impressive results in mathematics and coding. However, most approaches rely on static datasets, which have been suggested to encourage memorisation and limit generalisation. We introduce DÉJÀQ, a framework that departs from this paradigm by jointly evolving a diverse set of synthetic mathematical problems alongside model training. This evolutionary process adapts to the model's ability throughout training, optimising problems for learnability. We propose two LLM-driven mutation strategies in which the model itself mutates the training data, either by altering contextual details or by directly modifying problem structure. We find that the model can generate novel and meaningful problems, and that these LLM-driven mutations improve RL training. We analyse key aspects of DÉJÀQ, including the validity of generated problems and computational overhead. Our results underscore the potential of dynamically evolving training data to enhance mathematical reasoning and indicate broader applicability, which we will support by open-sourcing our code.

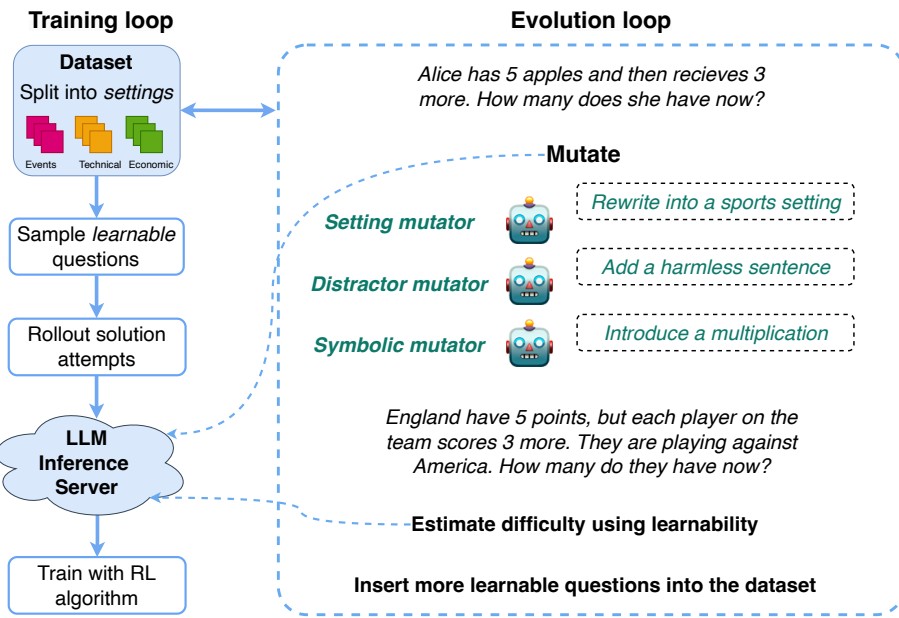

Figure 1: Overview of DÉJÀQ. We maintain an archive of problem-answer pairs, organised by the setting each question applies to. Training data for RLVR is sampled from this archive, which is continuously updated through various *mutators*. The *setting mutator* changes the setting (e.g., from Personal Life to Events), the *distractor mutator* introduces irrelevant information, and the *symbolic mutator* alters the underlying mathematical structure. Each problem is scored by its *learnability* and retained or replaced accordingly.

## 1 INTRODUCTION

Post-training of large language models (LLMs) is a highly active area of research, with recent methods focusing on designing training recipes that leverage real or synthetically generated datasets to enhance instruction-following ability (Ouyang et al., 2022; Wang et al., 2023b), coding performance (Nijkamp et al., 2023; Lozhkov et al., 2024), and mathematical reasoning (Shao et al., 2024; Hendrycks et al., 2021). Two key limitations are the scarcity of high-quality data and the substantial compute required for training. We approach both challenges through the following research question:

*How can we dynamically generate diverse and learnable training data that enables LLMs to bootstrap their own post-training?*

One of the central motivations for this question is the need to obtain training data that remains well-suited to the model's current capabilities. A commonly observed issue is the prevalence of training examples with (near-)zero variance, which provide little to no learning signal and introduce noise into gradient updates (Foster & Foerster, 2025; Yu et al., 2025). This not only hinders learning but also wastes valuable compute. Although such examples can be filtered manually, this only underscores the broader issue of limited and ineffective training data. In this work, we introduce DÉJÀQ, a method that evolves a dataset of challenging yet solvable problems, explicitly optimised to maximise the model's learning progress.

The design of DÉJÀQ builds on three complementary ideas that have proven effective in reinforcement learning, including to some extent LLM post-training. From ACCEL (Parker-Holder et al., 2022), we adopt the principle of evolving training data jointly with model optimisation, rather than relying on a fixed dataset. From RAINBOW TEAMING (Samvelyan et al., 2024), we incorporate the use of MAP-Elites to maintain a structured archive of diverse training problems, and apply LLM-guided mutations to generate new high-quality examples in sparsely populated regions of the search space. From *learnability-based training* (Foster & Foerster, 2025), we take *learnability* as a proxy metric for the expected utility of a datapoint during training. DÉJÀQ unifies these components into a single framework that evolves a dataset of verifiable problem-answer pairs through quality-diversity search for LLM post-training. The model continuously evaluates newly generated problems and retains those deemed sufficiently learnable, enabling open-ended bootstrapping without external supervision.

A core challenge in realising this framework is generating problems that are both *verifiable*, with ground-truth answers available by construction, and *skill-appropriate*, meaning they are neither trivial nor beyond the model's current capabilities. To address this, we introduce two complementary mutation strategies. The first is a curriculum-style approach that replaces problems with others expected to yield greater learning progress. The second is an LLM-guided strategy, in which the model rewrites existing problems either by modifying their contextual framing or by altering their structure in a controlled way. Structural changes include the insertion of distractors, which are semantically coherent sentences that do not affect the solution, as well as symbolic modifications to the underlying operations in the solution.

We evaluate DÉJÀQ using QWEN2.5-7B-INSTRUCT (Yang et al., 2024) on both in- and out-of-distribution mathematical problems. We find that combining curriculum learning with LLM-guided mutations yields substantially better performance than both standard RL fine-tuning and curriculum learning alone. We further evaluate how well the scoring function separates hard-but-solvable problems from flawed ones, measure how often mutators introduce such errors into the archive, and analyse the resource demands of the data-evolution pipeline. We summarise our main contributions below and provide a visual overview of our method in Fig. 1:

1. **DÉJÀQ - Synthetic Data Evolution:** An evolutionary framework for constructing a dataset of highly learnable, verifiable problem-answer pairs tailored to reasoning models.

2. **Simple but Effective Mutation Strategies:** We show that even simple LLM-guided mutators can effectively increase diversity and complexity while preserving verifiability, and we empirically compare the effectiveness of different mutation strategies.

3. **Efficient Bootstrapping:** The same model is used for both data generation and training, enabling a fully bootstrapped setup that leverages shared infrastructure.

4. **Empirical Validation:** We present a detailed empirical study showing that DÉJÀQ generates diverse and learnable problems for model training.

## 2 BACKGROUND

### 2.1 REINFORCEMENT LEARNING WITH VERIFIABLE REWARDS

Post-training of LLMs often involves a reinforcement learning (RL) phase, where a token-level Markov decision process (MDP) is defined by treating each token as an action and transitions as the concatenation of tokens to the existing context. Reinforcement Learning with Verifiable Rewards (RLVR) optimises the LLM using reward signals that can be automatically verified (Lambert et al., 2024). In mathematics, this may correspond to checking against ground-truth answers; in code generation, to evaluating against a test suite. Formally, RLVR maximises the objective,

$$\mathbb{E}_{y \sim \pi_\theta(x)} \left[ r_{\text{RLVR}}(x, y) - \beta D_{\text{KL}}(\pi_\theta(y \mid x) \parallel \pi_{\text{ref}}(y \mid x)) \right] \tag{1}$$

where $r_{\text{RLVR}}(x, y) \in \{0, 1\}$ denotes a verifiable binary reward, and the second term penalises deviation from a reference policy, weighted by the regularisation parameter $\beta$. Recently, the Group Relative Policy Optimisation (GRPO) algorithm has shown strong performance in mathematical domains (Shao et al., 2024). Unlike its predecessor, PPO (Schulman et al., 2017), GRPO avoids reliance on a learned value network by sampling multiple generations and estimating advantages directly from them, offering both simplicity and improved stability.

### 2.2 MAP-ELITES

To co-evolve a dataset of challenging yet solvable questions for the LLM to train on, we adopt a quality-diversity algorithm (Cully & Demiris, 2018), namely MAP-Elites (Mouret & Clune, 2015). MAP-Elites maintains an archive of items $x \in \mathcal{X}$, where each item is assigned a feature descriptor via a mapping $d : \mathcal{X} \to \mathbb{R}^n$, and scored by a fitness function $f : \mathcal{X} \to \mathbb{R}$. In our setting, the feature space is discretised into a finite grid by assuming that each dimension of $d(x)$ is categorical. The archive is initially populated with a set of seed items $\{x_1, \ldots, x_k\}$, each inserted into its corresponding cell. Thereafter, the algorithm proceeds iteratively: at each step, an item $x \in \mathcal{X}$ is sampled from the archive and modified by a mutation operator $q : \mathcal{X} \to \mathcal{X}$, yielding a new item $x' = q(x)$. The mutated item $x'$ is then assigned to a cell via $d(x')$, and scored using $f(x')$. Let $y$ denote the current occupant of that cell. If the cell is empty or if $f(x') > f(y)$, then $x'$ replaces $y$ in the archive. Through repeated application of this procedure, MAP-Elites constructs an archive that is both diverse and high-quality.

## 3 RELATED WORK

**Data curation**. At the core of this work is the idea that training on curated data is a more effective use of compute resources than uniform sampling. This idea is explicit in Unsupervised Environment Design (UED) (Dennis et al., 2020), where the environment distribution is adapted online so that agents encounter environments targeted at their current policy, for example by learning or evolving instances to maximise regret (Parker-Holder et al., 2022; Beukman et al., 2024) or by prioritising those with high value error in a replay buffer while stochastically injecting new ones for diversity (Jiang et al., 2021a;b). Related ideas appear in the intrinsic-motivation and curiosity literature, where learning progress quantifies improvements in prediction or goal achievement and is used to build automatic curricula (Oudeyer et al., 2007; Schmidhuber, 2010). In deep RL, learning-progress signals guide goal sampling or task selection so that agents focus on regions of the goal space where performance is changing most (Colas et al., 2019; Kanitscheider et al., 2021). More recently, LLM-guided open-ended exploration methods such as OMNI and OMNI-EPIC combine learning-progress-style curricula with foundation models that assess the interest or novelty of tasks (Zhang et al., 2024; Faldor et al., 2025).

In this work, we adopt *learnability* as the scoring criterion for data curation, which intuitively prioritises problems the model can solve but not yet consistently. For a given problem instance $x$ and model parameters $\theta$, it is defined as $l_\theta(x) = p_\theta(x)\big(1 - p_\theta(x)\big)$ (Tzannetos et al., 2023), where $p_\theta(x)$ denotes the probability that the model solves $x$ correctly. In contrast to regret-based criteria, it does not require estimating the return of an optimal policy or an upper bound on performance, which is particularly challenging in open-ended domains, and instead relies only on success probabilities under the current model (Rutherford et al., 2024).

**Curricula for LLMs**. Training large language models (LLMs) typically consists of two phases, pre-training and post-training, both of which require substantial data and compute. To maximise the utility of a fixed training budget, the design of effective learning curricula has emerged as a key strategy. In pre-training, Jin et al. (2023) introduce a sequence-length-based curriculum to improve efficiency, while Pouransari et al. (2024) apply a similar approach to address inefficiencies related to how documents are concatenated and chunked. Lin et al. (2024) propose Selective Language Modelling, which restricts loss computation to informative tokens. In post-training, recent state-of-the-art models have adopted hand-crafted curricula to guide training (Yu et al., 2025). Beyond manual design, adaptive curriculum learning has gained traction. Foster & Foerster (2025) propose upsampling examples with high learnability. Similarly, Xu et al. (2025) generate many on-policy rollouts but train only on the most informative samples. Qi et al. (2025) apply evolution to web-based LLM agents, progressively generating more complex tasks to drive continual improvement. Finally, Shi et al. (2025) propose selecting training samples based on their proximity to a dynamically determined target difficulty, encouraging the model to focus on examples that are neither too easy nor too hard.

**Reasoning models**. LLMs are increasingly deployed in domains such as mathematics and coding, driving the development of specialised reasoning models trained to solve complex problems via intermediate steps. Techniques like Chain-of-Thought (CoT) (Wei et al., 2022), Tree-of-Thought (ToT) (Yao et al., 2023) and Self-Consistency (Wang et al., 2023a) prompt models to articulate reasoning traces prior to producing answers. To further strengthen this capability, several iterative schemes have been proposed in which models generate reasoning samples, fine-tune on them, and repeat the process (Zelikman et al., 2022; Hosseini et al., 2024). More recently, reinforcement learning (RL) approaches have shown that effective reasoning strategies can emerge without explicit instruction (Shao et al., 2024; DeepSeek-AI et al., 2025). These methods often follow the inference-time compute paradigm, accepting increased computational cost during inference in exchange for improved downstream performance (Snell et al., 2024; Wu et al., 2025).

**Synthetic math problems**. Strong mathematical reasoning capabilities require high-quality training data, but such data is costly and difficult to curate at scale. As a result, synthetic data has emerged as a compelling alternative. MathScale (Tang et al., 2024) begins from a seed dataset, extracts key concepts, and instructs an LLM to recombine them into novel questions. PromptCOT (Zhao et al., 2025) follows a similar path, additionally transferring chain-of-thought rationales from existing problems to guide new generations. WizardMath (Luo et al., 2023) leverages GPT-4 to generate training data and supervise student models, outsourcing both tasks to a static external oracle, which inherently limits downstream performance. In work concurrent to ours, SPARQ (Havrilla et al., 2025) applies quality-diversity evolution to construct a training set scored by solve rate. Unlike our method, however, it performs only a single round of generation followed by supervised fine-tuning. While this restricts adaptivity, SPARQ demonstrates strong gains in out-of-distribution generalisation, though in-distribution improvements remain limited.

# 4 OPEN-ENDED EVOLUTION OF DIVERSE AND LEARNABLE VERIFIABLE PROBLEMS

Our objective is to evolve a dataset of highly learnable reasoning problems in tandem with model training, while preserving verifiability and diversity. To this end, we introduce DÉJÀQ, a post-training method that curates a stream of challenging yet solvable problems tailored to the model's current capabilities. Implementation details are provided in Appendix A.

## 4.1 TRAINING OVERVIEW

DÉJÀQ consists of two concurrent processes that operate on the same underlying model and share the same inference infrastructure: a data-evolution process and an RLVR training process. We refer to these as the *teacher* and the *student*, respectively.

**Dataset evolution**. Starting from a seed dataset used to initialise the archive, the teacher repeatedly selects a high-scoring parent together with a target category whose cell is underperforming. It then applies an LLM-guided mutation, queried through the shared inference server, that rewrites the parent

so that the resulting candidate belongs to the chosen category. The candidate is evaluated according to its *learnability* and inserted into the archive only if it outperforms the current occupant of that cell.

To estimate learnability, we approximate the success probability from $K$ completions generated by the shared inference server, yielding $\hat{p}_\theta(x)$ and the unbiased estimator $\hat{l}_\theta(x) = \frac{K}{K-1}\hat{p}_\theta(x)(1-\hat{p}_\theta(x))$. Malformed or unsolvable problems naturally receive low scores under this estimator, preventing them from persisting in the archive.

**RLVR training**. The training loop follows standard GRPO. At each step, the student samples a batch of problem–answer pairs from the archive, using a mixture of their estimated learnability (favouring higher values) and age (favouring more recent items). For each problem, the inference server produces multiple rollouts and rewards are computed from verifiable correctness with cosine scaling and format checks.

## 4.2 INITIAL ARCHIVE POPULATION

To instantiate the MAP-Elites archive, we require a seed dataset $\mathcal{D}_0$ and a descriptor function $d$ that maps each datapoint to a set of features or categories. For $\mathcal{D}_0$, we adopt all templates from GSM-Symbolic (Mirzadeh et al., 2024), a template-based variant of GSM8K (Cobbe et al., 2021) designed to mitigate overfitting in frontier models. While symbolic templates are not strictly necessary for our method, we leverage them to obtain a larger pool of high-quality seed data.

To define the descriptor function $d$, we manually inspect the templates and devise a classification scheme based on their *problem setting*, such as Professional, Economic, or Recreational. Each template is assigned a setting by instantiating a concrete example and prompting a language model (QWEN2.5-32B-INSTRUCT (Yang et al., 2024)) to generate a chain-of-thought rationale followed by a final classification. While we use a hand-crafted diversity axis, future work may explore automatically discovered descriptors (Bradley et al., 2024; Pourcel et al., 2024). The complete list of setting categories is provided in Appendix A, and the classification prompt is shown in Appendix D.

## 4.3 LLM-GUIDED MUTATIONS

Balancing expressivity with verifiability is a key consideration when constructing synthetic datasets for reasoning domains such as mathematics. Models require access to sufficiently challenging and diverse training data, yet the solutions to these problems must remain accessible to ensure meaningful supervision. Prior work circumvents this issue by relying on stronger teacher models to generate and validate data (Luo et al., 2023). To move beyond this dependence on external oracles, we introduce LLM-guided mutators that support continual self-improvement. Examples are shown in Fig. 2, and all prompts are detailed in Appendix D.

**Setting mutator**. We introduce an LLM-guided setting mutator inspired by Samvelyan et al. (2024). This mutator first identifies a category in the archive with low learnability and prompts an LLM to rewrite a high-quality parent problem to match that category. This enables exploration beyond the seed dataset, yielding more diverse and targeted problems. Crucially, the LLM is instructed to alter only the problem setting, leaving the reasoning structure and quantities unchanged, so the original solution remains valid.

**Distractor mutator**. Beyond contextual rewrites, we introduce a distractor mutator that adds semantically irrelevant sentences to a problem. These distractors provide additional detail or colour while preserving the original logic and solution.

**Symbolic mutator**. While the setting and distractor mutators change the presentation of the problem, the reasoning required to solve it is left unchanged. In contrast, the symbolic mutator modifies the mathematical structure of the problem and updates the solution accordingly. Since our model is trained to produce chain-of-thought reasoning, we prompt it to first propose an interesting modification and then solve the mutated problem step by step. This approach helps maintain both the correctness and diversity of the resulting examples.

## 4.4 PITFALLS OF EVOLUTION

---

**Base problem-answer pair and mutations**

A fog bank rolls in from the ocean to cover a city. It takes 256 minutes to cover every 9 miles of the city. If the city is 72 miles across from the oceanfront to the opposite inland edge, how many minutes will it take for the fog bank to cover the whole city?

**Setting:** Environmental  **Solution:** 2048

— **Setting Mutator** (Retain solution) —————————————————————

In a scientific experiment, a fog bank is generated to simulate atmospheric conditions. The fog bank travels at a consistent speed, taking 256 minutes to cover every 9 kilometers of the experimental field. If the experimental field is 72 kilometers across, how long will it take for the fog bank to completely cover the field?

**Setting:** Scientific  **Solution:** 2048

— **Distractor Mutator** (Retain solution) —————————————————————

A fog bank rolls in from the ocean to cover a city. It takes 256 minutes to cover every 9 miles of the city. The fog starts to move in from the sea, creeping over the rooftops slowly. If the city is 72 miles across from the oceanfront to the opposite inland edge, how many minutes will it take for the fog bank to cover the whole city?

**Setting:** Environmental  **Solution:** 2048

— **Symbolic Mutator** (Modify solution) —————————————————————

A fog bank rolls in from the ocean to cover a city. The fog bank's speed is 64 miles per 256 minutes at the start and decreases uniformly to half that speed by the time it reaches the end of the city, which is 72 miles across. How many minutes will it take for the fog bank to cover the whole city?

**Setting:** Environmental  **Solution:** 384

Figure 2: Example LLM-guided mutations of a fog coverage problem under the operators used in DÉJÀQ. Shown are real generations produced by the 7B base model and obtained using the same prompts as applied during training.

While LLM-guided mutations expand the space of candidate problems, they also introduce several practical challenges. We outline the main issues and the operational steps taken to address them.

**Avoiding overuse of heavily mutated parents.** Because each mutation is applied to an existing archive item, repeatedly selecting the same high-quality parents can reduce diversity and amplify errors. We track the mutation depth of each item and downweight the sampling probability of deeply mutated parents, ensuring that fresh or lightly mutated items remain competitive candidates.

**Preventing long-term error accumulation.** Mutation errors can propagate if flawed items repeatedly serve as parents. To counter this, we periodically refresh a small fraction of archive cells by replacing their occupants with clean problem–answer pairs drawn from the seed dataset. This ensures a steady influx of verifiable items and prevents the archive from drifting too far from a reliable base.

**Filtering for meaningful variation.** Trivial rewrites can pollute the archive without contributing new learning opportunities. Following RAINBOW TEAMING, we apply a BLEU-based filter (Papineni et al., 2002): a mutated candidate is only considered for insertion if its surface form differs sufficiently from the parent. This prevents minor paraphrases from entering the archive while keeping the mutation process lightweight.

**Maintaining relevance as the model improves.** As training progresses, items that were once challenging may become too easy. To avoid retaining stale problems, we apply exponential decay to stored learnability estimates and refresh them whenever the corresponding item appears in the GRPO batch. This ensures that the archive reflects the model's current abilities and promotes the continual discovery of hard-but-solvable problems.

Table 1: Mean accuracy with 95% confidence interval on QWEN2.5-7B-INSTRUCT. Bold indicates the best method per evaluation. Results for GPT-Eval-ID are omitted here as all models achieved accuracy above 95%; see Appendix C.2.

| Method | In-Distribution (%) | | | Out-of-Distribution (%) | |
|---|---|---|---|---|---|
| | Symbolic | P1 | P2 | MATH-500 | GPT-Eval-OOD |
| Base | $88.0 \pm 0.9$ | $77.4 \pm 1.2$ | $62.6 \pm 1.9$ | $68.0 \pm 4.1$ | $86.6 \pm 3.0$ |
| DR | $85.4 \pm 1.0$ | $63.4 \pm 1.3$ | $51.6 \pm 2.0$ | $62.6 \pm 4.2$ | $81.6 \pm 3.4$ |
| Resample | $87.6 \pm 0.9$ | $64.2 \pm 1.3$ | $46.4 \pm 2.0$ | $63.2 \pm 4.2$ | $79.6 \pm 3.5$ |
| DÉJÀQ-S | $\mathbf{94.1} \pm 0.7$ | $\mathbf{84.1} \pm 1.0$ | $64.4 \pm 1.9$ | $67.4 \pm 4.1$ | $86.6 \pm 3.0$ |
| DÉJÀQ-A | $94.1 \pm 0.7$ | $83.7 \pm 1.0$ | $\mathbf{65.5} \pm 1.9$ | $\mathbf{69.6} \pm 4.0$ | $\mathbf{89.0} \pm 2.7$ |

## 5 EXPERIMENTS

We experimentally evaluate DÉJÀQ using QWEN2.5-7B-INSTRUCT (Yang et al., 2024). Our code is implemented on top of TRL (von Werra et al., 2020) for RL fine-tuning on the LLMs and vLLM (Kwon et al., 2023) for the model inference server. All models are trained starting from the seed dataset of GSM-Symbolic templates (Mirzadeh et al., 2024). Details are provided in Appendix C.

**Methods**. As baselines, we include the original base model and RLVR with a *domain randomisation* (DR) strategy that uniformly samples from the set of available templates and instantiates them with valid parameters. We also consider a variant trained using the same evolutionary framework, but with mutations limited to resampling from the initial dataset. We compare these against two variants of DÉJÀQ: the *setting* mutator (DÉJÀQ-S), and the full combination of *setting, distractor, and symbolic* mutators (DÉJÀQ-A). We do not evaluate the distractor or symbolic mutators in isolation, as they cannot produce cross-category mutations.

**Benchmarks**. We evaluate mathematical reasoning on the Symbolic, P1, and P2 subsets of the GSM-Symbolic test set (Mirzadeh et al., 2024). The P1 and P2 suites can be regarded as progressively harder in-distribution variants, as they remain GSM questions but include one or two additional clauses that increase difficulty and move performance closer to an out-of-distribution regime. True out-of-distribution generalisation is assessed on MATH-500 (Hendrycks et al., 2021; Lightman et al., 2024). To isolate the contribution of open-ended LLM-guided mutations, we also construct two synthetic benchmarks with GPT-5. GPT-Eval-ID explicitly contains in-distribution GSM problems, while GPT-Eval-OOD features creative and varied out-of-distribution cases. Full construction details are provided in Appendix B.

### 5.1 INSIGHTS ON EVALUATION ACCURACY

Table 1 reports mean accuracy with 95% confidence intervals for the base model, the domain-randomisation (DR) baseline, the resample baseline, and the two DÉJÀQ variants. Results for GPT-Eval-ID are only provided in Appendix C.2, as all models achieved accuracy above 95%, indicating that performance on basic GSM questions is already saturated.

**In- vs. out-of-distribution performance**. Across all evaluations in Table 1, DÉJÀQ outperforms the baselines. On the most in-distribution tasks (Symbolic, P1), DÉJÀQ-S achieves the highest mean accuracy. This follows from its design, which increases surface-level variety while keeping the symbolic form unchanged, directly strengthening in-distribution performance. DÉJÀQ-A is a close second, showing that structural mutations do not significantly reduce in-distribution gains. On P2, DÉJÀQ-A becomes the best method. Repeatedly applying the structural mutations can recover a similar style of questions as in P2, since these mutations can also add two or more clauses to the base question. This makes them well-suited to handle the increased complexity of this benchmark. On clearly out-of-distribution tasks (MATH-500, GPT-Eval-OOD), DÉJÀQ-A also performs best. This supports the idea that combining setting, distractor, and symbolic mutations improves generalisation beyond the training distribution, consistent with the findings from SPARQ (Havrilla et al., 2025).

**Naive training degrades performance**. Both domain randomisation and resampling fail to improve over the base model and often reduce accuracy (e.g., P1: Base 77.40% vs. DR 63.42%; P2: Base

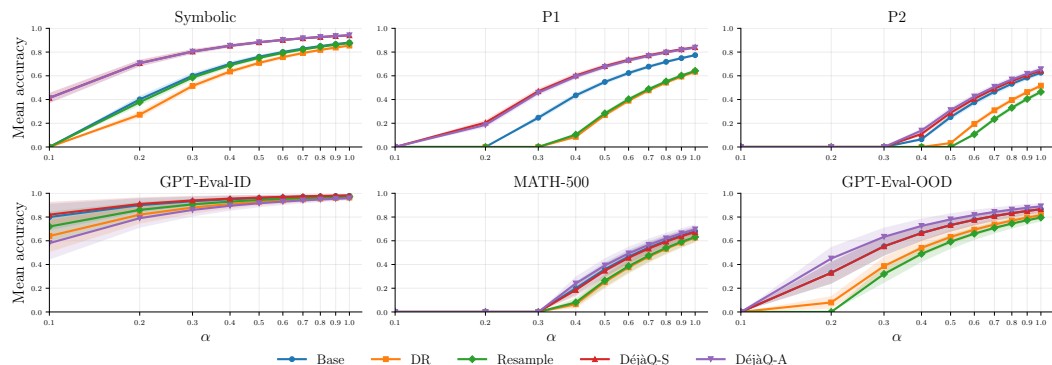

Figure 3: Mean accuracy under conditional value at risk (CVaR) across the six evaluation datasets. The x-axis denotes the risk parameter $\alpha$ (log scale), the y-axis shows mean accuracy, and shaded regions indicate 95% confidence intervals.

62.60% vs. Resample 46.44% in Table 1). The only cases where performance does not drop as sharply are the most in-distribution datasets (Symbolic and GPT-Eval-ID). A likely explanation is that the base model has already been post-trained on highly curated data, and further naive fine-tuning on comparatively basic distributions disrupts this carefully optimised state. By contrast, DÉJÀQ applies LLM-guided mutations that generate informative variation rather than indiscriminate training examples, which enables it to not only recover but surpass the base model's performance. These findings caution against unstructured post-training on generic data and support structured, learnability-driven data evolution as a safer and more effective path to robustness and generalisation.

**Robustness to challenging instances**. We evaluate robustness using Conditional Value at Risk (CVaR) (Rutherford et al., 2024), which measures the expected success rate over the hardest $\alpha$-fraction of tasks. For a given $\alpha \in (0, 1]$, CVaR computes the mean success on the lowest $\alpha$-percentile of task outcomes. Results for all datasets are shown in Fig. 3.

Across risk levels, both DÉJÀQ variants strictly dominate the baselines on most datasets and match them on the remainder. On Symbolic and GPT-Eval-ID, where overall accuracy is already very high, meaningful differences still appear for smaller $\alpha$, reflecting stronger tail robustness. The largest differences are observed on P1, P2, and GPT-Eval-OOD, where DÉJÀQ outperforms at every risk level and most clearly at lower $\alpha$, consistent with improved performance on the hardest instances. These results support the conclusion that learnability-driven selection combined with targeted LLM-guided mutations enhances tail performance and improves robustness both in- and out-of-distribution.

## 5.2 MAINTAINING VERIFIABILITY THROUGH LLM-GUIDED MUTATIONS

A key advantage of using RL to train LLMs for mathematical reasoning is the availability of ground-truth data. LLM-guided mutations risk undermining this by introducing errors into the training process. To assess this risk, we designed two controlled experiments, shown in Fig. 4.

**Setup**. The first experiment fixes the model to eliminate non-stationarity and then simulates the evolutionary pipeline by evolving the archive for 100 mutation rounds. We estimate learnability from 100 generations and decay it after each simulated sample call to emulate the GRPO callback. This setup captures the validity of questions produced during a realistic evolutionary process. In the second, we repeatedly mutated each of 200 seed problem-answer pairs along a linear chain of ten steps, without any evolutionary selection, to isolate the effect of mutation depth alone. In both cases, we perform these experiments on the base model QWEN2.5-7B-INSTRUCT with DÉJÀQ-S and DÉJÀQ-A as well as on the post-trained models with their respective mutator and use GPT-5-MINI as a reasoning oracle to estimate correctness for all problems generated over time.

**Learnability as a verifier**. The top row of Fig. 4 reports learnability versus invalidity. Base rates differ markedly across mutators: for the base model, DÉJÀQ-S yields $22.2\% \pm 2.9\%$, whereas DÉJÀQ-A yields $43.7\% \pm 3.4\%$. This gap is intuitive, as surface-level context rewrites are easier

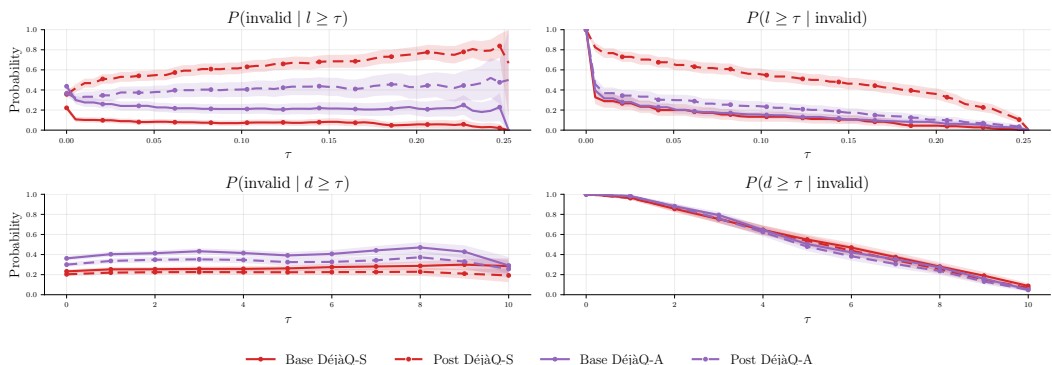

Figure 4: Estimated probabilities with 95% confidence intervals. The left column shows $P(\text{invalid} \mid x \geq \tau)$, i.e., the probability that a question is invalid given that its learnability or depth exceeds a threshold $\tau$. The right column shows the reverse conditional, $P(x \geq \tau \mid \text{invalid})$.

for an instruction-tuned base model than structural mutations that alter problem composition. After post-training, the rates shift to $35.7\% \pm 3.3\%$ for DÉJÀQ-S and $36.6\% \pm 3.4\%$ for DÉJÀQ-A. In other words, post-training raises the invalidity base rate for the *setting mutator* but lowers it for the *all mutator*. This suggests that improved student capabilities can feed back into the teacher, making mutations more reliable when variation spans multiple axes and supports out-of-distribution generalisation. In contrast, restricting the teacher to surface-level rewrites exhausts its benefit, as such variability cannot scale with the student's growing abilities.

Conditioned on invalidity, we observe that learnability decreases. When conditioning on learnability instead, base models show the expected pattern: the probability of invalidity declines as learnability rises, indicating that learnability acts as an effective filter. Post-trained models, however, exhibit the opposite trend, with high-learnability pairs being increasingly likely to be invalid. We conjecture that as the student becomes stronger, generating genuinely new and correct problems becomes increasingly difficult. As their share in the dataset declines, invalid problems occupy a larger fraction. Since our RLVR process optimises only the student's performance and never directly improves the teacher, this mismatch likely exacerbates the problem.

**Recursive mutations**. The bottom row of Fig. 4 shows that recursive application of mutators does not significantly increase the likelihood of invalid pairs. The conditional probability $P(\text{invalid} \mid d \geq \tau)$ remains stable across depths, with occasional dips at deeper levels due to early termination of chains after hard failures (e.g., JSON errors), which lowers measured invalidity among surviving items. The complementary curves $P(d \geq \tau \mid \text{invalid})$ decrease smoothly with $\tau$. Across both mutators, the post-trained model consistently yields fewer invalid generations than the base model. These results support the hypothesis from the learnability analysis: deeper mutation does not drive more errors, but rather improving the model's capabilities makes it harder to find genuinely new, correct problems. This underscores the need to *train the teacher* alongside the student so the mutator can keep pace with a stronger solver and continue generating diverse, verifiable problems.

## 5.3 RESOURCE ANALYSIS

In addition to serving the RLVR loop, the inference server is used for learnability estimation and LLM-guided mutations. It is therefore important to examine whether these extra calls introduce bottlenecks. Table 2 reports GPU and memory statistics across methods.

**Memory footprint and bandwidth**. Memory usage is stable across methods (about $65$–$74$ GiB, or $82$–$93\%$), showing that learnability estimation and LLM-guided mutations do not increase the model footprint. Memory utilisation rises with mutations (from $2.4\%$ for DR to $39.2\%$ for DÉJÀQ-A), but remains well below saturation, indicating that mutations mainly improve bandwidth usage rather than impose new constraints.

Table 2: Inference server GPU and memory statistics (mean $\pm$ standard deviation). *Memory (GiB)* and *Memory (%)* report allocated GPU memory; *GPU Util (%)* is the average streaming multiprocessor utilisation; *Mem Util (%)* is the average memory controller utilisation.

| Method | Memory (GiB) | Memory (%) | GPU Util (%) | Mem Util (%) |
|---|---|---|---|---|
| DR | $74.4 \pm 2.5$ | $93.4 \pm 3.2$ | $3.2 \pm 16.4$ | $2.4 \pm 12.5$ |
| Resample | $65.1 \pm 28.0$ | $81.7 \pm 35.2$ | $12.3 \pm 25.2$ | $3.6 \pm 7.6$ |
| DÉJÀQ-S | $71.6 \pm 7.4$ | $89.8 \pm 9.3$ | $21.2 \pm 36.3$ | $16.0 \pm 27.6$ |
| DÉJÀQ-A | $72.4 \pm 3.2$ | $91.0 \pm 4.1$ | $51.7 \pm 41.5$ | $39.2 \pm 31.8$ |

**GPU utilisation**. The DR baseline achieves very low utilisation ($3.2\%$), suggesting that training alone does not exploit the inference server efficiently. Learnability estimation (Resample) raises utilisation to $12.3\%$ and additionally performing a single round of LLM-guided mutations (DÉJÀQ-S) raises utilisation further to $21.2\%$. The full mutation pipeline (DÉJÀQ-A), which can chain up to three inference calls in one mutation reaches $51.7\%$ on average. High variance reflects bursty workloads rather than steady load. Thus, LLM-guided mutations make more effective use of available capacity without exhausting resources.

**Wall-clock effects and scheduling**. Due to high variability on the shared cluster, we do not report wall-clock comparisons. Nevertheless, runs with mutations were consistently slower. This likely stems from contention when training and evolution submit requests simultaneously: queues can delay training even if average utilisation is far from $100\%$. Lightweight scheduling, such as prioritising training queries or timing mutation requests to follow the completion of a training iteration, could alleviate these delays by better interleaving the two workloads.

## 6 CONCLUSION

We introduced DÉJÀQ, an evolutionary framework that leverages LLM-guided mutators to asynchronously evolve a dataset of diverse, learnable problems for reinforcement learning with verifiable rewards. Building on MAP-Elites, DÉJÀQ maintains an archive of synthetic problems, selecting and retaining those most learnable under the current model. Empirically, DÉJÀQ improves both in- and out-of-distribution performance and shows greater robustness to the most challenging instances. Our analysis demonstrates that increased mutation depth does not inflate failure rates and that DÉJÀQ requires only modest additional resources. Finally, while learnability is an effective proxy for verifiability in base models, post-trained models struggle to generate highly learnable valid samples. We hypothesise that this bottleneck arises from the teacher lagging behind the student, highlighting an important avenue for future work.

## REPRODUCIBILITY STATEMENT

The experimental setup is detailed in the main paper, with further specifications provided in Appendix C. All prompts used for LLM generations are included in Appendix D. Additional implementation details of DÉJÀQ are given in Appendix A. We will release our code and synthetic datasets publicly upon acceptance.

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

---

**Algorithm 1** The DÉJÀQ algorithm. Shared components are highlighted in blue.

---

**Require:** Initial model parameters $\theta_0$, seed dataset $\mathcal{D}_0$, mutation operator $q$, and training budget $T$
**Ensure:** A post-trained reasoning model with parameters $\theta_T$
1: Initialise LLM inference server
2: Initialise MAP-Elites archive $\mathcal{A} \leftarrow \emptyset$
3: Populate $\mathcal{A}$ with seed problems from $\mathcal{D}_0$ and compute learnability scores $l(x; \theta_0)$

4: **Launch two asynchronous processes:**
5: (1) Model Training Loop
6: **for** $t = 1$ to $T$ **do**
7:     Sample training batch $\mathcal{B}$ from $\mathcal{A}$
8:     Update model via RLVR:           ▷ Uses LLM inference server to sample generations

$$\theta_t \leftarrow \arg\max_\theta \mathbb{E}_{y \sim \pi_\theta(x)} \left[ r_{\text{RLVR}}(x, y) - \beta \, \mathrm{D}_{\text{KL}} \left( \pi_\theta(y \mid x) \,\|\, \pi_{\text{ref}}(y \mid x) \right) \right]$$

9: **end for**

10: (2) Dataset Evolution Loop
11: **while** training is running **do**
12:     Sample $x \sim \mathcal{A}$
13:     Generate mutant $x' \leftarrow q(x)$           ▷ Uses LLM inference server to propose mutations
14:     **if** $x'$ is correctly formatted **then**
15:         Compute score $s' \leftarrow l(x'; \theta_t)$     ▷ Uses LLM inference server to estimate learnability
16:         Assign descriptor $d \leftarrow d(x')$
17:         **if** $d \notin \mathcal{A}$ **or** $s' > l(\mathcal{A}[d]; \theta_t)$ **then**
18:             $\mathcal{A}[d] \leftarrow x'$
19:         **end if**
20:     **end if**
21: **end while**
22: **return** $\theta_T$

---

## A    DÉJÀQ IMPLEMENTATION DETAILS

In this section we provide all remaining implementation details for DÉJÀQ. Complete pseudocode is given in Algorithm 1.

### A.1    SETTING CATEGORISATION

In Table 3, we present the setting categorisation used in our DÉJÀQ experiments and was derived through a combination of manual analysis and LLM-assisted inspection.

### A.2    LLM INFERENCE SERVER INTEGRATION

Our approach integrates dataset curation into the same inference infrastructure used for training. We elaborate here on why the additional inference cost is justified and how this integration can be made efficient in practice.

First, prior work has shown that training on low-information samples can negatively impact model performance by slowing down overall training and introducing noise into the gradient updates (Yu et al., 2025; Foster & Foerster, 2025). Filtering out such instances in advance can therefore result in more effective gradient updates, offsetting the added inference cost. Second, recent RLVR methods employed in LLM post-training, such as GRPO (Shao et al., 2024) and VinePPO (Kazemnejad et al., 2024), already rely on fast, online sampling. These methods typically use a separate inference server such as vLLM (Kwon et al., 2023) to generate rollouts in real time. Importantly, this server is often underutilised during phases such as backpropagation or data staging.

By integrating dataset curation into the same inference infrastructure, we make more efficient use of available resources without incurring additional overhead. In our implementation, we employ an

Table 3: The setting categories used in our DÉJÀQ experiments.

| Name | Description |
|------|-------------|
| Personal Life | Scenarios from everyday personal experiences involving home life, family, school, food, health habits, or individual routines. |
| Professional | Contexts involving occupations, productivity, workplace responsibilities, or services rendered as part of a job or trade. |
| Economic | Situations involving money, costs, purchases, income, trade, markets, or financial decision-making. |
| Recreational | Scenarios focused on hobbies, play, sports, games, or other leisure activities pursued for enjoyment. |
| Events | Social or organised occasions such as birthdays, holidays, celebrations, school fairs, or community gatherings. |
| Scientific | Problems involving biological, chemical, or physical concepts, including natural processes and scientific observations. |
| Technical | Scenarios involving machines, devices, or engineered systems where understanding tools, parts, or operational constraints is essential. |
| Environmental | Scenarios involving ecosystems, weather, agriculture, conservation, or interactions between humans and the natural world. |

agnostic scheduling strategy that queries the inference server opportunistically, whether for training, scoring, or data generation. Identifying an optimal schedule that maximises throughput while avoiding interference with training remains a non-trivial engineering challenge and an open direction for future work.

### A.3 A LITTLE BIT TOO OPEN-ENDED?

We designed the featurisation of GSM-Symbolic templates to capture real-world domains we considered relevant. Because DÉJÀQ does not impose strict constraints on the types of problems generated, the model sometimes introduced unexpected axes of variation. For example, during development we observed smaller models rewriting problems from English into Spanish, occasionally mixing both languages while still producing valid math questions. Training on these examples does not improve performance on our current English-only benchmarks, but we hypothesise that it increases robustness along dimensions not measured by standard evaluations. This suggests the need for evaluation sets that better reflect the diversity and open-endedness of real-world problems, or, if the aim is to remain within a constrained domain, the use of auxiliary filtering mechanisms such as a judge model, as in RAINBOW TEAMING (Samvelyan et al., 2024).

## B GENERATING THE SYNTHETIC EVALUATION DATA

As outlined in Section 5, we construct two synthetic evaluation datasets using GPT-5 as the generator. These datasets are designed to assess the performance impact of DÉJÀQ under both in-distribution and out-of-distribution conditions. In total, we generated 500 problem-answer pairs for each dataset.

For the in-distribution dataset, we prompt GPT-5 with a description of the training distribution and request batches of 100 ideas. Each batch is balanced across a difficulty axis, with 30 easy, 40 medium, and 30 hard problems, and distributed evenly across settings. These ideas are then passed back to GPT-5 in a second prompt, which expands them into fully specified questions with corresponding answers.

For the out-of-distribution dataset, we instead encourage GPT-5 to propose maximally imaginative scenarios by varying both the settings (e.g., dreams, fantasy) and the narrative styles (e.g., diary entries, code blocks). Unlike the in-distribution case, we impose no constraints on the type of mathematics involved. The resulting ideas are subsequently transformed into complete questions and answers using a second query to the model.

The exact prompts used for dataset generation are provided in Appendix D.

## C  EXPERIMENT DETAILS

This section outlines the algorithms evaluated in our study and summarises the hyperparameters and computational resources used.

### C.1  ALGORITHMS

We summarise the algorithms evaluated in Section 5. All methods use the same RLVR process with GRPO and identical hyperparameters reported in Table 5; the only distinguishing factor is the distribution from which training data are sampled.

- **Domain randomisation.** Problems are sampled uniformly from the seed dataset, providing a simple reference point for comparison.

- **DÉJÀQ.** This method follows the standard RLVR pipeline but samples from an evolving archive according to *learnability*, rather than uniformly from a fixed seed set. The DÉJÀQ-S variant employs only the setting mutator, whereas DÉJÀQ-A uses the full suite of mutators (setting, distractor, and symbolic).

- **Resample.** This baseline mirrors the evolutionary loop of DÉJÀQ but replaces the mutation operator with a single procedure that resamples a new problem from the seed set. It ablates the contribution of LLM-guided mutations, testing whether improvements stem purely from learnability-based data selection rather than structured mutation.

### C.2  ADDITIONAL RESULTS

In Section 5 we omitted the results on the GPT-Eval-ID benchmark. We reproduce the full table of results in Table 4.

Table 4: Mean accuracy with 95% confidence interval on QWEN2.5-7B-INSTRUCT. Bold indicates the best method per evaluation.

| Method | In-Distribution (%) | | | | Out-of-Distribution (%) | |
|---|---|---|---|---|---|---|
| | Symbolic | P1 | P2 | GPT-Eval-ID | MATH-500 | GPT-Eval-OOD |
| Base | $88.0 \pm 0.9$ | $77.4 \pm 1.2$ | $62.6 \pm 1.9$ | $98.0 \pm 1.2$ | $68.0 \pm 4.1$ | $86.6 \pm 3.0$ |
| DR | $85.4 \pm 1.0$ | $63.4 \pm 1.3$ | $51.6 \pm 2.0$ | $96.4 \pm 1.6$ | $62.6 \pm 4.2$ | $81.6 \pm 3.4$ |
| Resample | $87.6 \pm 0.9$ | $64.2 \pm 1.3$ | $46.4 \pm 2.0$ | $97.2 \pm 1.4$ | $63.2 \pm 4.2$ | $79.6 \pm 3.5$ |
| DÉJÀQ-S | $\mathbf{94.1} \pm 0.7$ | $\mathbf{84.1} \pm 1.0$ | $64.4 \pm 1.9$ | $\mathbf{98.2} \pm 1.2$ | $67.4 \pm 4.1$ | $86.6 \pm 3.0$ |
| DÉJÀQ-A | $94.1 \pm 0.7$ | $83.7 \pm 1.0$ | $\mathbf{65.5} \pm 1.9$ | $95.8 \pm 1.8$ | $\mathbf{69.6} \pm 4.0$ | $\mathbf{89.0} \pm 2.7$ |

### C.3  COMPUTATIONAL RESOURCES

Experiments were run on a compute cluster with NVIDIA A40 and NVIDIA L40S GPUs (48 GB VRAM each). Every run used five GPUs: one dedicated to vLLM inference and four allocated to training.

Table 5: Combined Configuration Parameters for *training*, and *evolution*.

| Parameter | Value |
|---|---|
| *Training Parameters* | |
| reward_funcs | cos_correctness, format |
| reward_weights | 2.0, 1.0 |
| algorithm | GRPO |
| learning_rate | 1.0e-06 |
| lr_scheduler_type | cosine_with_min_lr |
| lr_scheduler_kwargs | min_lr_rate: 0.1 |
| gradient_accumulation_steps | 8 |
| gradient_checkpointing | true |
| gradient_checkpointing_kwargs | use_reentrant: false |
| num_generations | 6 |
| scale_rewards | true |
| max_prompt_length | 512 |
| max_completion_length | 2048 |
| per_device_train/eval_batch_size | 6 / 6 |
| num_iterations | 1 |
| max_steps | 500 |
| use_vllm | true |
| *Evolution Parameters* | |
| cell_size | 4 |
| ignore_top_k | 6 |
| score_decay | 0.95 |
| score_alpha | 0.5 |
| bleu_threshold | 0.6 |
| resample_prob | 0.25 |
| structure_probs | distractor: 0.4, symbolic: 0.4, both: 0.2, none: 0.0 |
| max_tries | 5 |
| mutation_batch_size | 8 |

## D  PROMPTS

---

**Qwen Math System Prompt**

Please reason step by step, and put your final answer within `\boxed{}`.

---

**Teacher System Prompt**

You are a knowledgeable and patient mathematics teacher. Aim to develop the student's intuition and problem-solving skills. You will be given math problems along with specific instructions, and your task is to revise or adapt the problems to best meet those instructions.

---

**Setting Mutator Prompt Template**

You will receive:
- **Candidate context**: A target setting for the problem, e.g., "Personal life".
- **Word problem**: A mathematical word problem.

TASK

Rewrite the problem to fit the *candidate context*. The story should clearly reflect this setting.

REQUIREMENTS

1. Preserve the mathematical structure and all quantities.
2. Change contextual details (names, objects, setting) to reflect the new context.
3. The result must be natural, coherent, and in English.

OUTPUT FORMAT

Start with a short reasoning:
- What is the original context?
- What changes will you make?
- What stays the same?

Then output a JSON:

```
{
    "mutated_problem": "<rewritten problem>"
}
```

INPUTS

Candidate context: {{ candidate_context }}
Word problem: {{ word_problem }}

---

## Distractor Mutator Prompt Template

You will receive:
- **Word problem**: A mathematical word problem.

TASK

Add a single harmless sentence that brings detail or colour, without changing the logic or answer.

REQUIREMENTS

1. Do not change the reasoning or introduce new relevant variables.
2. You may refer to quantities or names already present.
3. The result must be natural, coherent, and in English.

OUTPUT FORMAT

Start with a short justification:
- What sentence will you insert?
- Why does it not affect the answer?

Then output a JSON:

```
{
    "mutated_problem": "<problem with inserted sentence>"
}
```

INPUT

Word problem: {{ word_problem }}

---

## Symbolic Mutator Prompt Template

You will receive:
- **Word problem**: A mathematical word problem.
- **Solution**: The solution to the problem.

TASK

Make a meaningful change to the mathematical reasoning needed to solve the problem while ensuring the solution is updated accordingly. The goal is to create a new problem with a different solution, while keeping the rewrite as local and natural as possible.

REQUIREMENTS

1. Any change must be logically integrated into the story and affect the reasoning in a coherent way.
2. Preserve the original setting as much as possible.
3. The new problem must be solvable, consistent, clearly worded and in English.

OUTPUT FORMAT

Start with a short reasoning:
- Why is the current solution correct?
- What reasoning change are you making?
- How will you adapt the story?

Then output a JSON:

```
{
   "mutated_problem": "<rewritten problem>",
   "mutated_reasoning": "<step-by-step reasoning to solve the new
                         problem>",
   "mutated_solution": "$<new solution in LaTeX>$"
}
```

INPUTS

Word problem: {{ word_problem }}
Solution: {{ solution }}

---

## GSM Ideas Prompt

You are a dataset generator for **grade-school mathematics (GSM)** word problem *ideas* designed to mirror the training distribution.

TRAINING DISTRIBUTION RECAP

- Contexts to use: **Personal Life, Professional, Economic, Recreational, Events, Scientific, Technical, Environmental**.
- Problems should reflect everyday scenarios and straightforward wording.
- They must be solvable with only the four basic operations once instantiated.

TASK

Generate **100 creative problem ideas** with strong variety and balanced coverage of the eight contexts.

DIFFICULTY MIX (MANDATORY)

Produce **30 easy**, **40 medium**, and **30 hard** ideas.

- **Easy (30)**: 1–2 steps; small integers; simple narrative.
- **Medium (40)**: 2–3 steps; multiple operations; slightly more detail.
- **Hard (30)**: 3–5 steps; multi-stage reasoning (combine counts, change, simple averages/per-item pricing).

IDEA REQUIREMENTS

- Do **not** write full math problems or include answers.
- Instead, describe what the problem should look like: specify the scenario, objects involved, and type of operations required.
- Keep each description concise but concrete enough to later be turned into a full problem.
- Ensure variety of contexts and problem structures.

OUTPUT

Return a **JSON array** of 100 objects, each with keys:

- `"setting"`: the word-problem setting (string)
- `"problem"`: a short description of the intended problem (string)
- `"difficulty"`: one of `"easy"`, `"medium"`, or `"hard"`

Contexts (use exactly):

```
Personal Life, Professional, Economic, Recreational, Events,
Scientific, Technical, Environmental
```

Example skeleton:

```
[
  {
    "setting": "Personal Life",
    "problem": "A child sharing candies equally with friends.",
    "difficulty": "easy"
  },
  {
    "setting": "Economic",
    "problem": "A shopkeeper selling items at different prices and
                calculating total revenue.",
    "difficulty": "medium" }
]
```

Output only this JSON object—no additional text.

## GSM Generator Prompt

You are a dataset instantiator for grade-school math problems.
Input: a JSON array of ideas with keys {"setting", "problem", "difficulty"}.
Task: For each idea, write a full, self-contained word problem with clean numbers and matching the given difficulty. Provide the exact numeric answer.
Output: the same array but with keys {"setting", "problem", "answer"}.
Return only the JSON.

## Out-of-Distribution Ideas Prompt

You are a **maximally imaginative** dataset generator tasked with stress-testing a math model's ability to handle **open-ended** situations.

### CREATIVE LATITUDE
- Invent fantastical realms, speculative technologies, dream sequences, metaphors, or fourth-wall breaks.
- Vary narrative devices: dialogue snippets, diary entries, riddles, recipes, classified ads, stage directions, code comments, etc.
- Settings should feel strange, playful, or surreal—well beyond ordinary grade-school math.

### TASK
Generate **100 creative problem ideas** across three difficulty levels:
- **30 easy**
- **40 medium**
- **30 hard**

### IDEA REQUIREMENTS
- Do **not** write full word problems or provide answers.
- Instead, for each entry, provide a **short but concrete description** of what the math problem *could* look like.
- Vary not only the settings and narrative devices, but also the **style of mathematical reasoning** involved (e.g., arithmetic, geometry, probability, logic, algebra, combinatorics, number patterns).
- Strive for breadth and variety so that different kinds of mathematical thinking are represented across the collection.
- Be bold and diverse: embrace the fantastical, absurd, or stylistically unconventional.

OUTPUT

Return a **JSON array** of 100 objects, each with keys:

- `"setting"`: the broad setting (string)
- `"problem"`: a **creative description** of what the intended math problem should look like (string)
- `"difficulty"`: one of `"easy"`, `"medium"`, or `"hard"`

EXAMPLE SKELETON

```
[
    {
        "difficulty": "easy",
        "setting": "A dream entered in a journal",
        "problem": "..."
    },
    {
        "difficulty": "medium",
        "setting": "A time traveler shopping at a futuristic market",
        "problem": "..."
    },
    {
        "difficulty": "hard",
        "setting": "A stage play",
        "problem": "..."
    }
]
```

Output only this JSON object—no additional commentary.

## Out-Of-Distribution Generator Prompt

You are a maximally imaginative dataset instantiator tasked with stress-testing a math model's ability to handle **open-ended** situations.

INPUT

A JSON array of ideas, each with keys: `"setting": "..."`, `"problem": "..."`, `"difficulty": "..."`

TASK

For each idea, transform it into a **single, comprehensive, and natural-sounding math question**. The question should:

- Seamlessly weave the **setting** and **problem details** into one flowing narrative.
- Sound like a **standalone word problem**.
- Be phrased in **imaginative and engaging language**, but still precise enough that the math is well-defined.
- Match the requested **difficulty level**.

Then, provide a single **LaTeX-formatted expression** as the answer.

OUTPUT

Return a JSON array with keys:

- `"problem"`: the full, natural-sounding math question.
- `"solution"`: a step-by-step reasoning to solve the problem (string).
- `"answer"`: the LaTeX expression.

Return only the JSON.

