# OpenReview forum: "DéjàQ: Open-Ended Evolution of Diverse, Learnable and Verifiable Problems"
_ICLR.cc/2026/Conference — Submitted to ICLR 2026_

### Official Review · Reviewer_oZJP · 2025-10-16

**Soundness:** 4
**Presentation:** 3
**Contribution:** 3
**Rating:** 8
**Confidence:** 4

**Summary:**

This paper introduces DéjàQ, a novel framework for improving the mathematical reasoning abilities of Large Language Models (LLMs) through dynamic dataset evolution. Instead of relying on static datasets, DéjàQ co-evolves a diverse set of synthetic problems alongside model training.

The core of the contribution lies in three LLM-guided mutation operators that generate new problems: a "setting mutator" that alters the problem's narrative context, a "distractor mutator" that adds irrelevant information, and a "symbolic mutator" that modifies the underlying mathematical structure.

Crucially, the same model being trained is used to perform these mutations, creating an efficient, self-bootstrapping system. The paper also provides a thoughtful analysis of the validity of generated problems and the computational overhead of the system.

**Strengths:**

- **Novel Framework**: The framework is a novel and well-designed synthesis of evolutionary algorithms, curriculum learning, and self-improvement for LLM post-training.

- **Strong Empirical Performance**: The method demonstrates significant and consistent improvements over well-chosen baselines on a variety of mathematical reasoning benchmarks, including both in-distribution and out-of-distribution tasks.

- **Efficient Bootstrapping**: The use of the same model for both training and data generation is a key strength, eliminating the common reliance on more powerful external "teacher" models and making the system more self-contained and efficient.

- **Thorough Analysis**: The paper includes a high-quality analysis of the method's potential failure modes, including the verifiability of generated problems over time (Section 5.2) and the practical resource implications (Section 5.3).

- **Clarity of Presentation**: The paper is written with clarity, and the figures, particularly the system overview in Figure 1, are highly effective at conveying the core ideas.

**Weaknesses:**

- **Unresolved Teacher-Student Lag**: The paper identifies a key limitation where the "teacher" does not improve with the "student", leading to a higher proportion of invalid problems among high-learnability candidates after training. While identifying this is a strength of the analysis, the paper frames it as future work and does not propose or test a mechanism to resolve it. This might limit the truly "open-ended" nature of the evolution in the long run without further modification.

- **Scalability Questions**: The experiments are conducted on a 7B model. While the results are excellent, it remains an open question how the dynamics of this tightly coupled system would scale to much larger models (e.g., 100B+ parameters). For instance, the quality of mutations might improve, but the rate of student improvement might also accelerate, potentially exacerbating the teacher-student lag.

- **Dependence on Initial Seed Data**: The evolutionary process begins from a seed set of problem templates (GSM-Symbolic). While the mutations, especially the symbolic one, introduce significant novelty, the framework may still be fundamentally constrained by the mathematical concepts present in the initial seed data. It is unclear if the system could evolve problems requiring entirely new types of reasoning not represented in the seed set.

**Questions:**

- The symbolic mutator is arguably the most powerful operator. Could you provide a more detailed breakdown of its typical failure modes? For instance, beyond producing an incorrect final answer, how often does it generate problems that are logically inconsistent, ambiguous, or unsolvable?

- The analysis in Appendix A.3 mentions the model spontaneously generating problems in Spanish, which highlights the open-ended nature of the system. Did you observe any other surprising emergent behaviors? Specifically, did the symbolic mutator ever introduce mathematical operations or concepts that were not present in the original GSM-Symbolic templates, thereby increasing the conceptual complexity of the archive?

- In your related work, you cite Rainbow Teaming for its use of MAP-Elites and an archive to generate diverse problems. Have you considered citing OMNI-EPIC? It seems highly relevant and arguably closer to your work compared to Rainbow Teaming, as it also describes an open-ended evolutionary process that maintains an archive of generated tasks to create a curriculum of increasing difficulty.

---

> ### Author Response · Authors · 2025-11-14
> **Response to reviewer**
>
> We sincerely thank the reviewer for their supportive and thoughtful assessment, and for recognising the **novelty, empirical strength, and clarity of DéjàQ**.
>
> **The teacher–student lag**
> --
> We agree that this is an important limitation. However, such lag is standard in related work, which either rely on fixed teachers or alternate between discrete rounds of data generation and training. While we agree that it is a limitation, we believe that by *identifying and quantifying* this phenomenon in Section 5.2, we also provided valuable new insights.
>
> **Scalability**
> --
> We refer to the general comment for practical constraints. While our experiments focus on 7B models, we expect mutation quality to improve with model scale. As the reviewer rightly points out, this may also amplify the teacher–student lag, which we identify as an important direction for future work.
>
> **Seed dataset dependence**
> --
> We fully agree that the system’s expressiveness depends on the initial seed corpus. In future work, we plan to explore starting archives generated from alternative sources (e.g., GPT-5 or broader symbolic datasets) to encourage more diverse reasoning styles. In this work, we intentionally focused on the specific and narrow space of GSM problems to isolate the algorithmic effects of learnability-based sampling and LLM-guided mutation without confounding factors from domain variability.
>
> **Responses to specific questions**
> --
> 1. Symbolic-mutator failures are currently grouped under a single “invalid” label covering logical inconsistencies, unsolvable problems, and related errors, which we also observed in practice. A finer-grained categorisation could provide additional insight and is an interesting direction for future work.
> 2. Beyond the Spanish examples noted in Appendix A.3, we observed emergent use of *new mathematical operations* (e.g., reasoning with percentages) and increased compositional depth, showing the system’s capacity for open-ended expansion.
> 3. We thank the reviewer for drawing our attention to *OMNI-EPIC*, which indeed shares conceptual parallels with DéjàQ. We have now cited and discussed it in the related-work section.
>
> We greatly appreciate the reviewer’s positive assessment and hope that, given these clarifications and improvements, they might help advocate for our paper during the discussion phase.

---

> ### Comment · Reviewer_oZJP · 2025-11-25
>
> I thank the authors for their detailed and constructive response. I remain convinced of the paper's quality, novelty, and strong empirical results. I will maintain my score.

---

### Official Review · Reviewer_XXEE · 2025-10-31

**Soundness:** 3
**Presentation:** 4
**Contribution:** 2
**Rating:** 4
**Confidence:** 3

**Summary:**

This paper proposes DÉJÀQ, a self-bootstrapping framework that simultaneously evolves a curriculum of synthetic math problems and trains a 7B LLM via RL with verifiable rewards. Using MAP-Elites, it maintains a diverse archive of problem-answer pairs indexed by human-defined settings; three in-model mutators (setting, distractor, symbolic) continually rewrite problems, while an estimated learnability score filters offspring for neither-too-easy-nor-impossible instances. The same 7B model serves as both rollout model and mutation "teacher", yielding gains on GSM-Symbolic and MATH-500 without external labels or larger teachers.

**Strengths:**

* The paper introduces a fully self-contained framework that co-evolves a curriculum of synthetic math problems and improves a solver without any external oracle or larger teacher. By uniting MAP-Elites, learnability-based filtering, and three LLM-driven mutators inside a single RLVR loop, it removes the usual dependency on hand-written templates or proprietary generators, which is promising for community adoption.

* Experiments show consistent gains over benchmarks: +6–7 % accuracy on GSM-Symbolic subsets and +1.6 % on MATH-500, together with better tail robustness (CVaR).

* Resource measurements prove the extra inference calls fit within the idle time of the existing rollout server, validating practical deployability.

**Weaknesses:**

A substantive assessment of the weaknesses of the paper. Focus on constructive and actionable insights on how the work could improve towards its stated goals. Be specific, avoid generic remarks. For example, if you believe the contribution lacks novelty, provide references and an explanation as evidence; if you believe experiments are insufficient, explain why and exactly what is missing, etc.

* After post-training, high-learnability pairs being increasingly likely to be invalid (Fig. 4), indicating the teacher can no longer invent genuinely new, correct problems as the student becomes stronger.

* About "Verifiable": symbolic mutator lets the same model rewrite the question and supply the new chain-of-thought + ground-truth answer; there seems no adequate method to guarantee the validity or correctness.

**Questions:**

1. In Section 5.2, the paper mentioned: "Since our RLVR process optimises only the student's performance and leaves the teacher static, this mismatch likely exacerbates the problem." What exactly are the student and teacher in RLVR? Is it not the same model that trains and generates data? Why is the teacher static?

2. What reward is used in RLVR? During data generation, is format checking required and should a format reward be used to guide the process?

3. Section 4.4 lists many tricks without detailed explanations-could additional operational descriptions be added? After learnability scores decay, are high-learnability problems replenished during training? Otherwise, how is it ensured that problems are not selected repetitively while still keeping high-learnability problems for training?

4. The paper mentioned: "We do not evaluate the distractor or symbolic mutators in isolation, as they cannot produce cross-category mutations." In fact, only the symbolic mutator changes the problem answer and requires altering the computation steps in the output-why is the comparison focused only on whether the category changes?

5. Why do the experimental results show improvement on symbolic tasks, yet on GPT-Eval-ID the performance of DÉJÀQ-A worsens while only DÉJÀQ-S improves, even though the invalidity of new problems generated by DÉJÀQ-S increases with training?

6. The GSM dataset contains relatively simple problems. Could scalability and generality be issues for this method? The paper only conducts experiments on a 7B model; would smaller models struggle to generate high-quality questions and answers, while larger models can already solve most problems in the GSM dataset? Moreover, among the three mutation methods, none is designed to alter the difficulty level of the problems.

7. The experimental baselines are too few; there is no comparison with performances of other same-scale open-source models.

8. The classification prompt is NOT found in Appendix E.

---

> ### Author Response · Authors · 2025-11-14
> **Response to reviewer**
>
> We thank the reviewer for for highlighting that DéjàQ is a **self-contained, verifiable, and practically deployable framework** that removes dependence on external teachers.
>
> **The identified weaknesses**
> --
> We agree that the observed increase in invalidity among high-learnability pairs reveals a key limitation. Importantly, however, our analysis in Section 5.2 *makes this visible*. We consider this transparency a strength, as it isolates the source of degradation (teacher–student lag) and informs future extensions.
>
> **Responses to specific questions**
> --
> 1. The reviewer is correct in pointing out that we did not introduce this terminology. The *student* is the RLVR-optimised policy, while the *teacher* is the same model but prompted to generate new data instances. We clarified this in the revised version in Section 4.1. When we say that the teacher remains static, we meant that it is never explicitly trained to become a better teacher. We changed the wording in the relevant paragraph to better reflect this.
> 2. The reward combines a *cosine correctness term* and a *format-compliance term*, following [1]. This is now stated directly in Section 4.1.
> 3. We thank the reviewer for highlighting this section, which has been reworked in the updated version. Learnability decay gives every problem a finite sampling lifespan and prevents pathological questions from being selected forever. For example, a question with two equally valid answers that always split model predictions and therefore keeps $p(1-p)$ high. Without decay, such items would never leave the archive and would repeatedly dominate sampling. We do not explicitly replenish high-learnability items: new problems enter with fresh learnability and naturally replace older, decayed ones, so diversity is maintained through turnover rather than preserving any item indefinitely.
> 4. We follow *Rainbow Teaming*’s evolutionary setup, where mutators must be capable of cross-category transitions. This follows from the fact that a parent from one category needs to be mutated to a candidate in another category. This rationale is now clearly explained in Section 4.1.
> 5. GPT-Eval-ID is essentially saturated, and the observed differences are not statistically significant. We hypothesise that DéjàQ-S exhibits a higher invalidity rate because surface-level rewrites do not sufficiently challenge a well-trained model. As the archive replaces questions only when they are more learnable, invalid examples are more likely to persist once they reach this threshold.
> 6. During the development phase of this work, we observed that smaller models struggle to sustain meaningful mutations. In general, the base model must already possess basic competence in the target domain (for 7B, GSM-level reasoning). While the mutators do not directly adjust difficulty, the *learnability-based selection pressure* naturally favours tasks that are increasingly challenging.
> 7. We acknowledge that broader baseline coverage would be desirable; however, we prioritised isolating algorithmic contributions within resource constraints (see general comment).
> 8. This was an oversight on our part. The missing classification prompt has been added to Appendix E.
>
> We thank the reviewer for these constructive points. We believe the revised version is now significantly clearer and better contextualised, and we hope this will be reflected in the final assessment.
>
> **References**
> --
> [1] Dang, Q.-A., & Ngo, C. (2025). Reinforcement learning for reasoning in small llms: What works and what doesn’t. https://arxiv.org/abs/2503.16219

---

> ### Author Response · Authors · 2025-11-26
> **Follow-up on reviewer comments**
>
> Thank you again for your detailed review. We would kindly appreciate it if you could consider our previous response. We have addressed all of your comments in detail and would be happy to engage further if anything remains unclear. For convenience, we summarise below how we revised the paper in response to your main points:
>
> - We clarified the teacher–student terminology and updated Section 4.1 to reflect this more clearly.
> - We added an explicit explanation of the reward function, including both correctness and format-compliance terms.
> - We substantially revised the discussion of learnability decay and clarified how it prevents pathological dominance while maintaining diversity.
> - We explained why mutators must be capable of cross-category transitions, following Rainbow Teaming.
> - We clarified why differences on GPT-Eval-ID are not statistically significant and why DéjàQ-S tends to retain more invalid cases.
> - We explained competence requirements for smaller models, the role of learnability-based selection pressure in driving difficulty, and constraints on additional baselines.
> - We added the previously missing classification prompt to Appendix E.
>
> We hope you will find that the paper has improved and will adjust your score accordingly.

---

### Official Review · Reviewer_ZWDX · 2025-11-01

**Soundness:** 2
**Presentation:** 2
**Contribution:** 2
**Rating:** 2
**Confidence:** 4

**Summary:**

The paper introduces DejaQ, a framework for open-ended evolution of synthetic training data in reasoning domains, particularly mathematical problem solving. Instead of relying on static datasets, DejaQ co-evolves problem–answer pairs alongside model training using LLM-driven mutations (setting, distractor, symbolic). These mutations aim to increase dataset diversity and adapt difficulty to the model’s current capability. The authors integrate this approach with RLVR and MAP-Elites to manage diversity and learnability. Experiments with QWEN2.5-7B-INSTRUCT show performance gains over standard RL baselines and domain randomisation, especially in robustness and out-of-distribution generalisation.

**Strengths:**

- Interesting integration of evolutionary search and RL-based fine-tuning for dataset generation.
- Demonstrates a novel use of LLM-guided mutations that preserve verifiability while diversifying data.
- Evidence that DejaQ improves robustness and OOD performance.

**Weaknesses:**

- Should also show results on a different family of LLMs (e.g., llama) instead of just Qwen. Different families of LLMs might have different behaviours.
- Another ablation of mutating the samples but not having the learning progress sampling would be useful to see which components contribute to the algorithm's overall performance.
- The authors assume that GPT-5-mini is a "reasonable" oracle. A better scientific practice would be to show on a dataset or human annotations on how good GPT-5-mini is as a judge.
- For the result of post-training and invalidity base rate, it would be interesting to see the same plot (fig 4) for each mutation separately. Since "post-training raises the invalidity base rate for the setting mutator but lowers it for the all mutator", seeing the changes for each type of mutation could give more insight into the differences between DejaQ-A and DejaQ-S.

- Since defining learning progress is a key part of the paper, it is missing a lot literature on learning progress.

The concept of learning progress in prediction or curiosity-driven networks originates from Schmidhuber’s early work on artificial curiosity in 1991 (see historical overview in https://ieeexplore.ieee.org/stamp/stamp.jsp?arnumber=5508364). It was later formalized by Oudeyer and Kaplan in 2007 as a computational mechanism for intrinsic motivation (https://ieeexplore.ieee.org/stamp/stamp.jsp?arnumber=4141061, https://pmc.ncbi.nlm.nih.gov/articles/PMC2533589/). Around 2013, Oudeyer’s group introduced the notion of competence progress, measuring improvement in goal achievement or task completion to drive exploration and skill acquisition (https://www.sciencedirect.com/science/article/pii/S0921889012000644 and related works from 2013–2014). Since 2018, this principle has been integrated into intrinsically motivated deep reinforcement learning frameworks (https://arxiv.org/abs/1810.06284, https://arxiv.org/abs/1906.08190). More recently, similar approaches have been applied in complex environments such as Minecraft (https://arxiv.org/pdf/2106.14876) and in LLM-guided data generation settings https://arxiv.org/abs/2306.01711).

**Questions:**

- The behaviour descriptors are handcrafted (i.e., manually inspected by the authors to come up with the templates). Could this part be potentially automated? e.g., approaches like QDAIF (https://arxiv.org/abs/2310.13032) or ACES (https://arxiv.org/abs/2310.10692)
- As with Goodhart's law, when a measure becomes a target, it ceases to be a good target. Did they authors see any pathologies happening when optimizing for the proposed learning progress metric? Discussion on how this issue could be solved would be useful.
- It is said that the "initial learnability become stale as the model improves", and so the "learnability scores are decayed over time". It would be useful to see an ablation whereby the learnability scores are recalculated, to see how much of this is a problem.
- The authors show an ablation of keeping the same evolutionary process but resampling from the initial dataset. Is the resampling based on the learning progress metric in this baseline?
- How do the authors know quantitatively/ qualitatively if a task set is out-of-distribution?
- What is the "risk parameter alpha"?

---

> ### Author Response · Authors · 2025-11-14
> **Response to reviewer**
>
> We appreciate the reviewer’s careful reading and for recognising the **novel integration of evolutionary search with RL-based fine-tuning** and the **improvements in robustness and OOD generalisation**.
>
> **Regarding model families**
> --
> As noted in the general comment, scaling experiments across multiple model classes would incur prohibitive computational costs. We deliberately focused on Qwen2.5-7B-Instruct as a strong, representative, open-source model to isolate algorithmic effects rather than model-specific differences.
>
> **Mutator ablations**
> --
> We agree that more granular ablations would provide additional insight. Our current analysis of mutation chains already partially isolates the impact of each mutator by examining how their applications correlate with invalidity rates. However, each mutator is itself a composition of several atomic edits (e.g., DéjàQ-S alters settings; DéjàQ-A additionally introduces distractors and symbolic-solution changes). Fully disentangling these atomic components would require redefining the operators and rerunning the entire evolutionary training pipeline so the archive reflects those finer-grained interventions—a computationally expensive procedure whose conclusions may also be highly domain-specific. Our goal in this work is therefore to evaluate the regime-level contributions of the broader mutation classes rather than optimise each atomic edit in isolation.
>
> **Oracle selection**
> --
> We agree that the ideal way to assess judge quality is through human annotation. However, our analysis required validation of many thousands of questions, and full human validation is prohibitively expensive at this scale. Additionally, since the oracle differs from all trained models, any residual noise is uncorrelated and does not affect *relative* performance comparisons.
>
> **Learning-progress literature**
> --
> We thank the reviewer for highlighting these relevant papers. We have added a thorough discussion of different data curation methods, including the connection to intrinsic motivation, to the related work section.
>
> **Responses to specific questions**
> --
> 1. We agree that automating this aspect is an exciting direction. We now mention this explicitly in Section 4.2.
> 2. Appendix A.3 discusses the observed pathologies. In particular, smaller models occasionally generated mathematically sound questions in Spanish, reflecting optimisation along an unintended axis. We mitigated this through prompt adaptation. When prompting alone is insufficient, auxiliary filtering mechanisms such as a judge model, as used in Rainbow Teaming [1], can be applied.
> 4. We note that this is not first observed by us, but originally in [2]. Re-estimating learnability continuously would indeed be informative but is computationally intensive, requiring repeated inference passes.
> 5. The resample baseline applies the same evolutionary process as DéjàQ, but its sole “mutator” resamples a new problem from the seed set instead of generating one. This baseline identifies whether improvement arises purely from learnability-based data selection rather than structured mutation.
> 6. We define in-distribution tasks as those conforming to GSM-Symbolic templates; deviations from these (format or reasoning style) are treated as out-of-distribution.
> 7. The “risk parameter” $\alpha$ corresponds to the quantile used in *Conditional Value-at-Risk* (CVaR), which measures expected success on the hardest $\alpha$-fraction of tasks, thereby capturing robustness to difficult cases.
>
> We are grateful for this detailed and constructive feedback, which has significantly improved both the literature framing and methodological clarity. We hope the reviewer finds these additions satisfactory and will reconsider the score accordingly.
>
> **References**
> --
> [1] Samvelyan, M., Raparthy, S. C., Lupu, A., Hambro, E., Markosyan, A. H., Bhatt, M., Mao, Y., Jiang, M., Parker-Holder, J., Foerster, J., Rocktäschel, T., & Raileanu, R. (2024). Rainbow teaming: Open-ended generation of diverse adversarial prompts. In Advances in neural information processing systems 38: Annual conference on neural information processing systems 2024.
>
> [2] Parker-Holder, J., Jiang, M., Dennis, M., Samvelyan, M., Foerster, J. N., Grefenstette, E., & Rocktäschel, T. (2022). Evolving curricula with regret-based environment design. In International conference on machine learning, ICML 2022.

---

> ### Author Response · Authors · 2025-11-26
> **Follow-up on reviewer comments**
>
> Thank you again for your detailed review. We would kindly appreciate it if you could consider our previous response. We have addressed all of your comments in detail and would be happy to engage further if anything remains unclear. For convenience, we summarise below how we revised the paper in response to your main points:
>
> - We added new related-work discussion linking DéjàQ to learning-progress and intrinsic-motivation-based data curation.
> - We expanded the discussion on mutator ablations and explained why finer-grained, atomic ablations would require reconstructing the entire evolutionary pipeline.
> - We clarified the choice to restrict experiments to Qwen2.5-7B-Instruct due to the computational cost of scaling across multiple model families.
> - We justified the oracle choice and clarified why any residual noise does not affect relative comparisons.
> - We addressed all specific questions.
>
> We hope you will find that the paper has improved and will adjust your score accordingly.

---

### Official Review · Reviewer_gWP3 · 2025-11-08

**Soundness:** 2
**Presentation:** 3
**Contribution:** 3
**Rating:** 4
**Confidence:** 4

**Summary:**

The paper introduces DéjàQ, a system for RLVR post-training where the dataset is evolved by a curriculum as well as LLM mutation. Evaluation results with Qwen2.5-7B-Instruct show the merits of this approach.

**Strengths:**

1. The paper presents a novel application of LLM mutators (to RLVR post-training)
2. The fact that the dataset evolution process can reuse the same inference infrastructure makes the approach more practical.
3. There are detailed analyses on robustness, maintaining verifiability, and resource requirements and hardware utilization.
4. The paper is well-written.

**Weaknesses:**

1. The paper would benefit from a clearer or more detailed description of the training process of DéjàQ and baselines. See Questions 2 and 3 below. (For example, the paper currently reads as though the training set and test set could’ve been identical (or at least the set of GSM-Symbolic templates used is identical across the training set and test set), and clarification from the authors would be appreciated.)

2. The fact that the RLVR baselines result in *worse* performance than the base model suggests that they were not implemented/engineered/tuned properly, since properly implemented RLVR should not result in worse performance. If that is indeed the case, then comparison with these baselines is not meaningful.

**Questions:**

1. Line 300 says “We do not evaluate the distractor or symbolic mutators in isolation, as they cannot produce cross-category mutations.” Why does the inability to produce cross-category mutations justify not evaluating distractor/symbolic mutators in isolation?

2. Could you explain more about how the seed training set is generated? Lines 192-193 say that the seed training set is GSM-Symbolic, but I’m not aware of an explicit training split for GSM-Symbolic. (https://huggingface.co/datasets/apple/GSM-Symbolic only contains a test split.)

3. Can you describe the “resampling” baseline in more detail? The current explanation (line 297) is not very clear to me.

4. In Algorithm 1, how often is the dataset evolved? In other words, how many iterations of (2) occur for every iteration of (1)?

5. Some of the benchmarks (GPT-Eval-ID, GPT-Eval-OOD) were generated by an LLM. How was it ensured that these synthetically generated benchmarks are of high quality (e.g., are error-free)?

---

> ### Author Response · Authors · 2025-11-14
> **Response to reviewer**
>
> We thank the reviewer for appreciating the **practicality of our design**, the **depth of our analyses**, and the **clarity of presentation**.
>
> **Clarifications and updates**
> --
> We have introduced Section 4.1 and Appendix C.1 detailing the full training procedures for DéjàQ and all baselines. In short:
> - The **domain randomisation** baseline uses uniform sampling from the seed dataset.
> - **DéjàQ** follows the standard RLVR pipeline but samples data from an evolving archive according to *learnability*, rather than uniformly from a fixed seed.
> - The **resample** baseline applies the same evolutionary process as DéjàQ, but its sole “mutator” resamples a new problem from the seed set instead of generating one. This baseline identifies whether improvement arises purely from learnability-based data selection rather than structured mutation.
>
> **RLVR baseline performance**
> --
> We agree that the lower RLVR scores appear surprising. All baselines were implemented with the same hyperparameters that were recommended in related work [1]. The only factor that differs is the *data distribution*. We hypothesise that continued training for carefully optimised models on the relatively easy GSM-Symbolic data may reduce performance, especially out of distribution.
>
> **Responses to specific questions**
> --
> 1. The evolutionary loop, adapted from *Rainbow Teaming*, generates candidates by applying a mutator that can map between categories. Briefly, questions are selected from one category and the LLM is instructed to change this into a new question belonging to a different category. Mutators unable to perform cross-category transitions (e.g. symbolic or distractor in isolation) are therefore not evaluated independently. We have clarified this rationale in Section 4.1.
> 2. We generated the training set using exclusively the symbolic split from the GSM-Symbolic [2]. The [supplementary repo]([google.com](https://github.com/apple/ml-gsm-symbolic/tree/main/templates/symbolic)) provides JSON templates with parameterized placeholders and variable-level constraints. We compiled 90 of the 100 templates into Python code and used rejection sampling to enforce the specified constraints. Ten templates were excluded because their constraints had extremely low satisfaction probability, making generation impractical.
> 3. The resampling baseline simply draws new problems by sampling from these 90 Python generators according to their natural template distribution: i.e., each sample is a fresh instantiation of an existing template, with no mutation or modification applied.
> 4. Dataset evolution and training run concurrently, and the ratio between number of evolutionary steps per training step is determined purely by their relative runtimes. In practice, for the all-mutator configuration, the system produces roughly 5–10 evolutionary steps per training step, but this is an empirical consequence of runtime differences, not a fixed schedule.
> 5. All synthetic benchmarks (GPT-Eval-ID/OOD) were generated with GPT-5-Mini as an oracle. Manual inspection of random samples revealed no errors. Moreover, since the oracle differs from all trained models, any residual noise is uncorrelated and does not affect *relative* performance comparisons.
>
> We thank the reviewer for these detailed and helpful questions, which have strengthened the paper’s clarity. We hope these clarifications address the concerns and encourage an upward revision of the score.
>
> **References**
> --
> [1] Dang, Q.-A., & Ngo, C. (2025). Reinforcement learning for reasoning in small llms: What works and what doesn’t. https://arxiv.org/abs/2503.16219
>
> [2] Mirzadeh, S.-I., Alizadeh, K., Shahrokhi, H., Tuzel, O., Bengio, S., & Farajtabar, M. (2024). GSM-symbolic: Understanding the limitations of mathematical reasoning in large language models. https://doi.org/10.48550/ARXIV.2410.05229

---

> ### Author Response · Authors · 2025-11-26
> **Follow-up on reviewer comments**
>
> Thank you again for your detailed review. We would kindly appreciate it if you could consider our previous response. We have addressed all of your comments in detail and would be happy to engage further if anything remains unclear. For convenience, we summarise below how we revised the paper in response to your main points:
>
> - We clarified all training procedures by adding Section 4.1 and Appendix C.1, including how DéjàQ, domain randomisation, and the resample baseline differ.
> - We explained the unexpectedly lower RLVR performance and why data distribution, rather than hyperparameters, is the primary factor.
> - We detailed how mutators perform cross-category transformations and why certain mutators are not evaluated independently.
> - We described how GSM-Symbolic templates were processed.
> - We explained the concurrency between dataset evolution and training, and why the effective ratio of evolutionary steps is runtime-determined.
> - We clarified the evaluation setup using LLMs as an oracle.
>
> We hope you will find that the paper has improved and will adjust your score accordingly.

---

### Author Response · Authors · 2025-11-14
**General Comment**

We sincerely thank all reviewers for their thoughtful feedback and for recognising the potential of DéjàQ. We are encouraged that every reviewer independently highlighted the **practicality**, **robustness improvements**, and **clarity of presentation** of our work. For ease of review, all **revisions and newly added text are highlighted in blue** throughout the manuscript.

In particular:
- All reviewers agree that our approach offers a **promising direction for scaling RLVR in LLMs**, effectively leveraging the *same inference infrastructure*.
- All acknowledge that the paper is **well written and well motivated**, with remaining clarifications now addressed in the revised manuscript.
- There is broad consensus that **DéjàQ substantially improves robustness and out-of-distribution generalisation**, confirming its value beyond standard RLVR post-training.

**Alternative models**
--
We acknowledge the request for experiments across multiple model families and sizes. However, such experiments would require rerunning the entire RLVR and curriculum-evolution pipeline, including all ablations, which exceeds our current resource budget. We therefore focused on a single strong open model (Qwen2.5-7B-Instruct) to ensure representativeness and reproducibility. Our goal is to analyse the algorithmic contributions such as learnability-based sampling, LLM-guided mutation, and evolutionary curriculum design, rather than architectural or scaling effects, which we view as complementary future work.

**Summary of main changes**
--
- Added a subsection to the related-work section discussing prior work on methods for data curation (Section 3).
- Introduced Section 4.1 to provide a clearer overview of the training process of DéjàQ and to explain how the learnability-based scoring function and shared inference infrastructure are integrated.
- Updated Section 4.4 to provide additional context on the stabilisation techniques applied to the evolutionary process.
- Included a detailed summary of all baselines in Appendix C.1.


We hope the clarifications and improvements provided in this revision will help reviewers converge towards a positive consensus.

---

### Meta-Review · Area_Chair_yT6q · 2026-01-02

**Summary:**

I recommend rejection.

The paper presents a novel, well-executed framework with strong empirical results.

However, despite the rebuttal effectively addressing clarity issues, significant concerns remain:

(1) experiments are limited to single QWEN2.5-7B-INSTRUCT model without different scales or cross-family validation, (2) a teacher-student lag is identified and but no solution is proposed, and (3) limited ablation studies. The authors decline to conduct addtional experiments due to computational cost, leaving these issues unresolved and introducing substantial uncertainty. Overall, the unresolved concerns outweigh the strengths, leading to rejection.

**Reviewer Concerns:**

**Largely addressed:**
1. Training procedure clarity and baseline definitions (gWP3, XXEE)
> Authors added provide details of the training process and baselines in Sec 4.1 and Appendix C.1.

2. Specific technical questions (all reviewers)
> Authors provided clear answers about resampling baseline, dataset generation, cross-category mutations, reward functions, and missing prompts

3. Missing related work on learning progress and intrinsic motivation (ZWDX)
> Authors added a discussion on related work regarding learning progress and intrinsic motivation.

**Outstanding:**
1. Limited model diversity (ZWDX, oZJP)
> Authors explicitly declined to run experiments on other model families (like Llama) due to resource constraints. While understandable for a 7B model training run, the lack of cross-model validation remains a significant empirical gap.

2. Teacher-student lag (oZJP, XXEE)
> Authors acknowledged that the "teacher" remains static and that invalidity rates rise at the end of training. They frame this as a strength of their transparency and a subject for future work, but do not propose or test a solution.

3. Lack of fine-grained mutator ablations (ZWDX)
> Authors explain why atomic-level ablations are expensive and decline to conduct additional experiments

4. Underperforming RLVR baselines (gWP3)
> Authors acknowledge it appears surprising and provide hypothesis about data distribution effects. This is plausible but not empirically validated.

5. Oracle Validation (ZWDX)
> The authors defended the GPT-5-mini oracle by stating that residual noise is uncorrelated with the trained models. However, they did not provide the human-annotated sample validation requested by ZWDX to prove the oracle's ground truth is actually correct.

**Reviewer Scores:**

- Reviewer oZJP (score 8): Explicitly stated the score would remain unchanged.

- Reviewer XXEE (score 4): Likely remain unchanged. Might increase slightly to 6.
> Clarity issues were fixed. The "teacher-student lag" was acknowledged as a known limitation.

- Reviewer ZWDX (score 2): Unlikely to change.
> Might raise to 4 given related work additions and clarifications. But core concerns about model diversity, oracle validation and missing ablations remain.

- Reviewer gWP3 (score 4): Likely increase to 6
> Rebuttal directly clarifies training details, seed generation, and baselines, addressing most questions, though baseline underperformance explanation might not fully convince.

---

### Decision · Program_Chairs · 2026-01-26

Reject